# Factors Influencing Weight Loss Practices in Italian Boxers: A Cluster Analysis

**DOI:** 10.3390/ijerph17238727

**Published:** 2020-11-24

**Authors:** Stefano Amatori, Oliver R. Barley, Erica Gobbi, Diego Vergoni, Attilio Carraro, Carlo Baldari, Laura Guidetti, Marco B. L. Rocchi, Fabrizio Perroni, Davide Sisti

**Affiliations:** 1Department of Biomolecular Sciences, University of Urbino Carlo Bo, 61027 Urbino, Italy; s.amatori1@campus.uniurb.it (S.A.); d.vergoni@campus.uniurb.it (D.V.); marco.rocchi@uniurb.it (M.B.L.R.); fabrizio.perroni@uniurb.it (F.P.); davide.sisti@uniurb.it (D.S.); 2Centre for Exercise and Sports Science Research, School of Medical and Health Sciences, Edith Cowan University, 6027 Joondalup, Australia; o.barley@ecu.edu.au; 3Faculty of Education, Free University of Bozen, 39100 Bozen, Italy; attilio.carraro@unibz.it; 4Faculty of Psychology, Uni E-Campus, 00182 Rome, Italy; carlo.baldari@uniecampus.it; 5Department of Movement, Human and Health Sciences, University of Rome “Foro Italico”, 00135 Rome, Italy; laura.guidetti@uniroma4.it

**Keywords:** boxing, cluster analysis, combat sports, professional athletes, weight cutting

## Abstract

It is common practice in combat sports that athletes rapidly lose body weight before a match, by applying different practices—some safer and others possibly dangerous. The factors behind the choice of practices utilised have not been fully studied. This study aimed to investigate the weight loss strategies used by Italian boxers and to look at the difference between higher and lower risk practice adaptors. A modified version of a validated questionnaire has been sent to 164 amateur (88%) and professional (12%) boxers by email. A heatmap with hierarchical clustering was used to explore the presence of subgroups. Weight loss strategies were used by 88% of the athletes. Two clusters were found, defined by the severity of weight loss behaviours. Professional fighters, high-level athletes and females were more represented in Cluster 2, the one with more severe weight-loss practices. These athletes were characterised by a higher weight loss magnitude and frequency throughout the season and reported being more influenced by physicians and nutritionists, compared with the boxers in Cluster 1. Not all the weight loss practices are used with the same frequency by all boxers. The level of the athlete and the boxing style have an influence on the weight-cutting practices.

## 1. Introduction

Combat sports have been rapidly growing in popularity over the past two decades, with ~25% of Olympic gold medals being in some form of combat sport, as well as professional sports (e.g., boxing, mixed martial arts and kickboxing) drawing millions of spectators [1]. In the 2021 Olympic Games, there will be six combat sports, with the new entry of karate. In combat sports, athlete body mass is verified at an official ‘weigh-in’, and they are then divided into categories by their body weight, in order to ensure competitors of similar size [2].

It is common practice for athletes to lose significant body weight in the days/weeks leading up to the competition to gain an advantage by being paired with a smaller opponent: this practice is colloquially referred to as ‘weight-cutting’. This weight-loss can be achieved by several different strategies, including energy intake restriction (gradual dieting and fasting), total body fluid reduction (restricting fluid intake, increasing sweat response (heated wrestling, plastic suits, saunas and spitting) and pseudo extreme/abusive medical practice (laxatives, diet pills, diuretics, enemas, sporting bulimia (vomiting)) [3,4]. Some of these could be dangerous for the athletes’ health: for example, moderate and severe dehydration increases the risk of acute cardiovascular problems, such as heart stroke and ischaemic heart disease. Furthermore, a severe dehydration could potentially increase the risk of brain injury due from head trauma induced by the strikes. Other potential risks related to weight-cutting practices include suppressed immune function, changes to insulin sensitivity and hormonal imbalances [2].

The use of weight-cutting has been observed to be highly prevalent (60–80%) across a wide range of combat sports including Brazilian jiu-jitsu, boxing, judo, mixed martial arts (MMA), muay-thai, kickboxing, taekwondo, wrestling and karate [3,4,5]. While the majority of athletes utilise gradual dieting and increased energy expenditure through exercise, there is a significant portion of athletes that report using more potentially dangerous methods (such as severe heat exposure), though the precise magnitude is unclear [3,4,5,6]. Furthermore, while previous research has reported more broad data on weight-cutting in combat sports, there is a paucity of data examining the potential differences in factors between athletes that choose to use more potentially dangerous methods of weight loss than those who focus on the less extreme methods. Factors such as competitive level, weight class and who influences their decision making may explain such choices. Developing a deeper understanding of weight-cutting in combat sports is essential, as the practice has been associated with negative health outcomes (including several deaths worldwide) [2,7] as well as impaired performance [3,8,9,10], although this is a topic of debate [11,12].

Boxing is one of the most popular combat sports worldwide. Statistics showed that, in 2017, in the United States the number of participants in boxing (of all ages and categories) exceeded six million people; in the UK in 2018 around eight hundred thousand were reported to be participating in boxing [13]. In boxing, research has failed to observe any positive or negative relationship between weight-cutting and competitive performance [12,14]. Despite no clear evidence that weight-cutting improves performance, the practice has been found to be highly prevalent, with research reporting >92% of all boxers to engage in some form of weight loss for competition [3,4]. However, the research has not conducted an in-depth exploration of the potential factors influencing athletes who chose to utilise higher-risk methods. Such an exploration is important, as it will allow organisers and regulatory bodies to better tailor communication to athletes and coaching staff when trying to manage extreme weight loss practices.

The primary aim of this cross-sectional study was to investigate the common weight-cutting strategies used by a sample of professional and amateur boxers competing in Italy. A secondary aim was to investigate if clusters existed regarding the strategies and the frequency of the rapid weight loss behaviours.

## 2. Materials and Methods

### 2.1. Participants and Study Design

A total of 164 subjects (144 males and 20 females) were recruited for this cross-sectional study. All participants were active boxers, both amateur (Olympic-style boxing) and professionals. They were recruited from several clubs in the period between July and November 2019; the Italian Boxing Federation, through the regional managers and coaches of the National team, helped to reach a bigger number of athletes. Participation was voluntary, and the data collected were completely anonymous; after a verbal explanation of the study, participants signed informed consent. For the forty-eight under 18 participants, consent to participate was signed by their legal guardians. The study was conducted in accordance with the Helsinki Declaration, and the approval to conduct the survey was received by the Ethics Committee of Urbino University (no approval number has been provided since it was not mandatory, given that no sensitive data were collected in the study).

### 2.2. Survey

A snowball sampling technique, often used in populations which are difficult for researchers to access, has been used in order to reach a larger number of athletes. Then, an online survey (Google Forms) has been sent via email to every participant who decided to take part in the study. The link to the survey has been sent to each participant by only one researcher, and the mailing list was protected by a password in order to not be accessible from others. The survey was based on the ‘Rapid Weight Loss’ questionnaire (RWLQ) of Artioli et al. [15], that has been already used in other similar studies [3,16,17]. It was composed of two sections: in the first one, personal data were collected, like age, gender, sports experience (level, category, number and level of matches) and weight-loss history (e.g., how many kilos were lost before a match, how many times weight was cut in one season). On the second section, subjects were asked to indicate—using a four-level Likert scale (0 = never done; 1 = almost never; 2 = sometimes; 3 = always)—if they applied each of the following 14 weight-cutting strategies (gradual weight-loss, increased training volume, water restriction, wear a plastic suit during the day/night, wear it during training, skip meals or fasting, to train in a heated environment, sauna, diet pills, diuretics, laxatives, to vomit, to spit saliva). The RWLQ provides a validated RWL score (RWLS) that allows a quantitative measure of the aggressiveness of the weight-cutting behaviours. On a latter section of the survey, not included in the original version of Artioli et al. [15], subjects were asked to indicate, on a three-level Likert scale (1 = no influence; 2 = moderate influence; 3 = big influence), in which extent the following figures may influence their behaviours regarding weight-cutting strategies: physician/nutritionist, master, physical coach, family, other boxers/teammates.

### 2.3. Data Analysis

The *pheatmap* R package [18] was used to create a heatmap of the 14 weight-cutting strategies, with the *euclidean* distance and *complete* method. Hierarchical clustering analysis was performed on the subjects, with the relative dendrogram. Elbow and silhouette methods were used for determining the optimal number of clusters, using the *fviz_nbclust* function of the *factoextra* R package [19]. A Permanova has been conducted using the function *adonis2* of the *vegan* R package [20] to check the statistical difference between the two clusters. A chi-square test has been used to test differences in weight-cutting strategies used and in participants characteristics between the two clusters, applying the false discovery rate correction (Benjamini–Hochberg) (*FSA* R package [21]). Results are presented with *p*(𝜒^2^) and Cramer’s V. V-values should be interpreted as >0.5 = high association, 0.3 to 0.5 = moderate association, 0.1 to 0.3 = low association, 0 to 0.1 = little or no association. A Mann–Whitney test has been used to test the difference in weight-cutting history and RWLS between the two clusters. All the analyses have been performed using SPSS 26.0 (IBM, Armonk, NY, USA) or R Studio 3.6.2 (RStudio PBC, Boston, MA, USA); GraphPad Prism 8 (GraphPad Software, San Diego, CA, USA) has been used to build figures.

## 3. Results

### 3.1. Participants Characteristics

A sample of 164 athletes of various levels took part in this survey. Of them, 144 (88%) were males and 20 (12%) were females, 48 (29%) participants were under 18, 145 (88%) were amateur athletes (it means that they compete in Olympic-style boxing) and 19 (12%) were professional boxers; 29 athletes (18%) included in this sample have won a medal in an international competition (tournaments, Worlds or Olympic Games). Athletes who had won or had medalled at an international competition were deemed “highest calibre,” those who won or had medalled at a national competition were deemed “moderate calibre,” and all others were classified “lesser calibre” for the analyses, as previously done by Reale et al. [16]. Of the 164 athletes, only 20 (12%) reported to have not ever applied acute weight loss strategies before a match. Detailed characteristics of the sample are reported in Table 1.

### 3.2. Clustering and Heatmap

A heatmap (Figure 1) was built with the reported frequencies of the 14 different weight-cutting strategies, by each of the 164 subjects. A two clusters solution has been found to be the best, using both Elbow and Silhouette methods; it can be graphically noted by the dendrogram. Cluster 1 comprises 133 subjects (81%), while the other 31 subjects (19%) belong to Cluster 2. The two clusters significantly differ one from the other (Permanova test; F_(1, 162)_ = 32.57, *p* < 0.0001).

In Table 2, the subjects’ responses to the specific questions on weight-cutting habits are reported, divided for the two clusters. 20 athletes (12.2%) reported no lifetime experience of weight loss before competition. As it can be noted, the two clusters differ for the maximal number of kilos loss before a single match in the entire career (*p* = 0.003), how many times a subject cut their weight in the previous season (*p* = 0.016), the average weight loss before a competition (*p* = 0.012), and the number of kilos usually regained in the week after the competition (*p* < 0.001); so, the severity of rapid weight loss behaviours is higher in the subjects pertaining to the Cluster 2.

In Figure 2, the frequency of use of the different strategies in the two clusters is reported. The two clusters significantly differ on all the strategies analysed (χ^2^ test; *p* < 0.05), except for the ‘Gradual Weight Loss’ (*p* = 0.529). It is interesting to note that, with the practices becoming more dangerous to health, the difference between the two clusters becomes more visible.

### 3.3. Rapid Weight Loss Score

The above-reported results (Table 2 and Figure 2) are well summarised by the ‘Rapid Weight Loss Score’ (RWLS), that was significantly higher in the Cluster 2, respect to the Cluster 1 (Mann–Whitney U = 302, *p* < 0.0001), as reported in Figure 3. The RWLS provides a measure of the severity of the weight-cutting behaviours. RWLS had a mean of 18.4 (±7.9) in the Cluster 1, and of 35.3 (±10.2) in the Cluster 2.

### 3.4. Socio-Demographic Differences between the Two Clusters

In Table 3, the socio-demographic differences between the two clusters are reported. This analysis has been selected in order to explore associations between some dependent variables, such as age, gender, education level, boxing category and level, and people of influence, and the two clusters as predictive factors. The variables which are dependent on the clusters are: gender, with a higher percentage of females pertaining to the Cluster 2; the category, with a higher prevalence of professional athletes in Cluster 2, in respect to the amateur ones; the level of the boxers, with higher calibre athletes (subjects who achieved a medal in an international competition) more present in Cluster 2. Regarding the people who might have an influence on choosing the weight-cutting strategies, only physicians/nutritionists seem to have an impact on these patterns, with the athletes in Cluster 1 reporting a lesser influence on their choices by these figures. It is worth noting that there is no difference in the age classes between the two groups.

## 4. Discussion

In this cross-sectional study, the common rapid weight loss strategies used by a sample of professional and amateur boxers were investigated and the presence of clusters have been explored relating to the severity (type and frequency) of each practice used. In this study, 88% of the participants reported having used rapid weight loss strategies during their careers. The strategies which can be applied to achieve a rapid weight loss are various, from the safer (e.g., gradual weight loss, increased training volume) to the most severe and potentially dangerous ones, including body water manipulation (fluid restriction, glycogen depletion, induced dehydration), sweating methods (both passive sweating, e.g., sauna and heated rooms, and exercise-induced sweating) and gut content manipulation (food restriction, use of laxatives, diuretics, vomiting) [2,4]. All these practices are not used with the same frequency by all boxers. Indeed, the main result of the study is the presence of two clusters of boxers in our sample, characterised by the severity of the weight-cutting practices they are used to apply. The cluster with more severe weight loss practices has a higher proportion of professional fighters, high-level athletes and females, and is also marked by a higher number of kilos usually lost before a match and regained in the week after that, and a higher number of times the subject cut the weight in the previous season.

Regarding the weight loss strategies, Reale and colleagues [16] reported that boxers favour the more gradual approaches to weight loss with respect to other combat sports. Indeed, the most used practices in our sample were the gradual weight-loss, increased training volume, and other practices linked to augmented sweat rate. These practices are the same reported to be most prevalent by Barley et al. [3]. Although infrequently reported, harmful practices such as vomiting and use of banned diuretics or laxatives remain concerning. The weight loss history and the magnitude of weight loss in our boxers are comparable with those reported in the study conducted by Barley et al. [3]. Indeed, the average weight loss before a competition and the weight regained in the week after the match are similar in the two studies, although slightly lower in the present sample; the boxers in our study were used to lose weight in almost half of the days before competition than those reported in the study by Barley et al. [3]. The effect of the duration of weight loss practices, its relationship with the severity of the practices used, and its consequences to subsequent weight regain, need to be further explored.

The heatmap (Figure 1) shows that the usage frequency of different weight-cutting strategies is not the same in all the subjects. Indeed, two clusters of subjects have been found in our sample. Cluster 1 is characterised by the use of gradual weight loss and increased training as the most prevalent strategies, while Cluster 2 had a greater presence of those strategies which can be referred to as gut content manipulation (use of laxatives, diuretics, vomiting and spitting saliva), that would provide a greater risk to athlete physical health [7]. The athletes in Cluster 2 had a greater magnitude of total weight loss compared with those from Cluster 1 (2.8 ± 2.2 kg and 1.7 ± 1.2 kg, respectively) and a higher frequency of weight cutting throughout a competitive season (Table 2). A greater amount of weight loss would plausibly result in an increased risk of impaired physical performance [3,9,22]. However, it is unclear if such reductions in physical performance are outweighed by the benefit of competing against smaller opponents; such complications may contribute to the lack of clear benefit or detriment on performance as a result of weight cutting [1]. These results confirm that there is a significant population of athletes within boxers that disproportionately use higher-risk strategies when trying to lose weight for competitions.

A higher proportion of females is present in Cluster 2, with respect to Cluster 1. To the authors’ knowledge, no studies have deeply investigated the differences in weight-cutting strategies between male and female fighters, and this should be an important topic for future research. It should be noted the absence of significant differences in the presence of under 18 athletes in the two clusters. This means that a good proportion of the youth athletes (about 22%) pertaining to Cluster 2 apply the same dangerous weight-loss practices as adults. The authors believe that this point deserves attention, as practices like manipulation of the gut content (use of diuretics, laxatives, or vomiting) represent a risk for the fighters wellbeing [2], in particular for youth. Future research should look to better understand the motivations of such athletes and their prevalence in other combat sports to help regulators and organisers when trying to design mitigation strategies for weight-cutting practices.

A higher proportion of top-level boxers (athletes who at least won or acquired a medal at an international competition) was present in the Cluster 2. A higher prevalence of extreme weight loss practices in higher calibre athletes has been previously reported by Reale et al. [16]. Higher calibre athletes apply more severe weight-cutting strategies, but it could be argued that they also have access to more support staff and thus would feel more comfortable to attempt difficult weight cuts. This is potentially confirmed by the greater reliance of Cluster 2 subjects on physicians and nutritionists (Table 3). Overall, the coach seems to have the biggest influence on the boxers’ choices in both clusters; this result is in accordance with those previously reported by Reale et al. [16], whilst Barley et al. [4] reported a high influence also of other boxers (partners or opponents). Another important difference between the two clusters is represented by the boxing category, as amateur/Olympic-style vs. professional boxing, with pro boxers that are more represented in Cluster 2. These two boxing styles follow different rules as per match duration, protective gear and scoring. The amateur boxing tournaments feature several matches over several days, unlike professional boxing where fighters have several weeks (and sometimes months) between bouts. Furthermore, professional boxing is also known as “prizefighting”, because matches are fought for a purse that is divided between the boxers as determined by contract, whilst amateur boxers cannot fight for money. Competing for a low number of times per year, and fighting-for-pay, professional boxers could be inclined to use more risky strategies to cut the weight before the matches. Furthermore, the time between weigh-in and the competition differs between amateur/Olympic-style and professional boxing. As for the Italian regulations, in amateurs, the weigh-in is the same day as the competition, while in professionals the day before [23]. A longer time between the weigh-in and the match could allow the athletes to apply more severe weight-loss strategies in order to lose more weight, having more time to recover and refuel before the match. It is worth noting that research has focused more on Olympic-style boxing, while professional boxing has been less investigated.

This study provides an overview of the weight-cutting strategies used both in amateur and professional boxing. These results could be useful for teams and Federations, in order to develop educational programs which could improve the knowledge and awareness of coaches and athletes regarding the risks involved with rapid weight loss and the recommended procedures to do that safely and adequately. The inclusion of physicians and nutritionists in the decision process regarding the weight loss strategies used could improve the safety of these procedures, reducing the health risks for the athletes. These findings also highlight the lack of information about weight-cutting practices in female and youth boxers, populations that surely need further consideration. However, the study presents some limitations: first of all, the use of an online survey to collect the data. Although this relies on self-reported data, a sufficiently large total athlete sample was recruited, and a highly validated combat sports survey was used in an attempt to mitigate this confounder. Additionally, the results are relative to the Italian context, and so the results generalisability cannot be taken for granted. Lastly, we did not collected data regarding the time between the weigh-in and the match, information that could have been useful for a better understanding of the phenomenon.

## 5. Conclusions

The cluster analysis put a focus on the characteristics of fighters who apply more severe practices (i.e., losing greater amounts of body weight as well as using methods such as fluid restriction, heat exposure and gut content manipulation), showing that professionals and higher-level fighters are more prevalent in Cluster 2. Fighters pertaining to this cluster also cut their weight more frequently during a competitive season, accompanied by a higher number of kilograms lost each time, and then regained to a greater extent after the match. Coaches are the most influential figures for the athletes, even if those in Cluster 2 are also conditioned by physicians and nutritionists. Possible future research could monitor a group of high-level athletes for the weeks before and after their matches, in order to have some reliable data on real behaviours and to avoid the use of potentially less reliable (self-reported) survey data.

## Figures and Tables

**Figure 1 ijerph-17-08727-f001:**
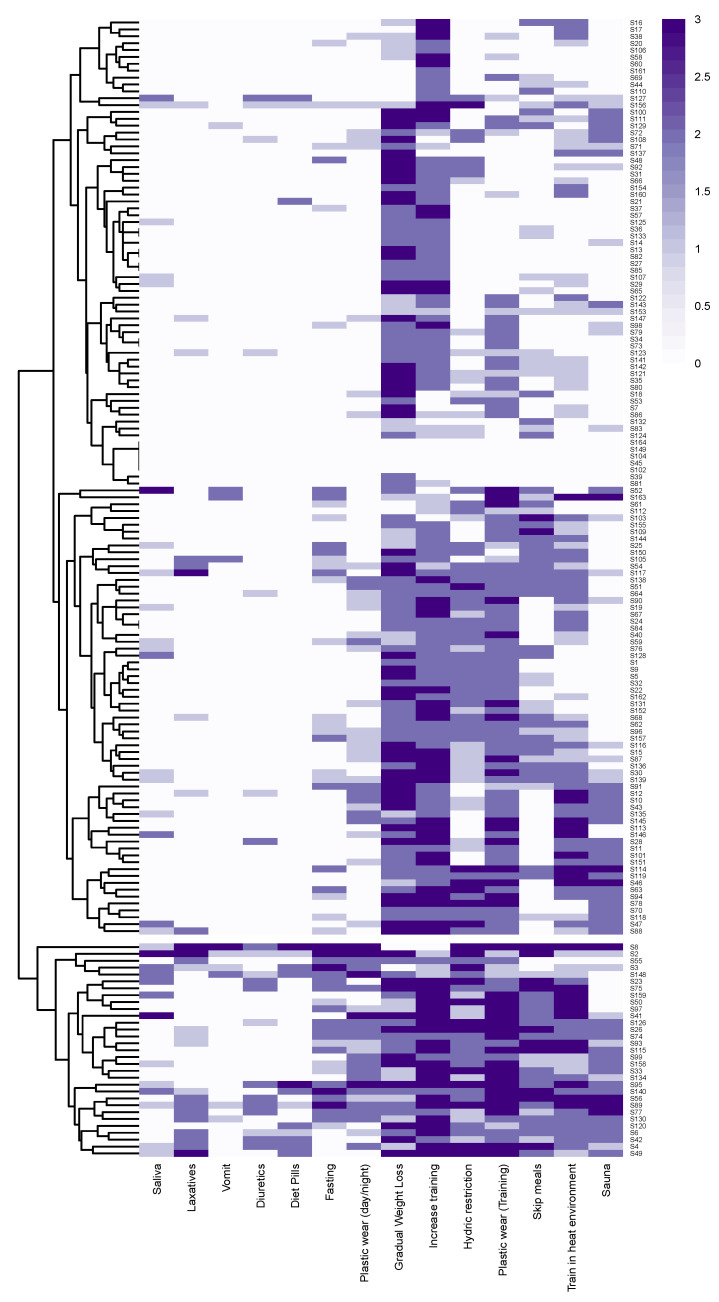
Heatmap with weight-cutting strategies (in columns) and subjects (in rows). An empty line separates the two clusters (Cluster 1 at the top, Cluster 2 at the bottom). White colour is associated with the ‘never used’ answer, while purple represents the ‘always’ answer.

**Figure 2 ijerph-17-08727-f002:**
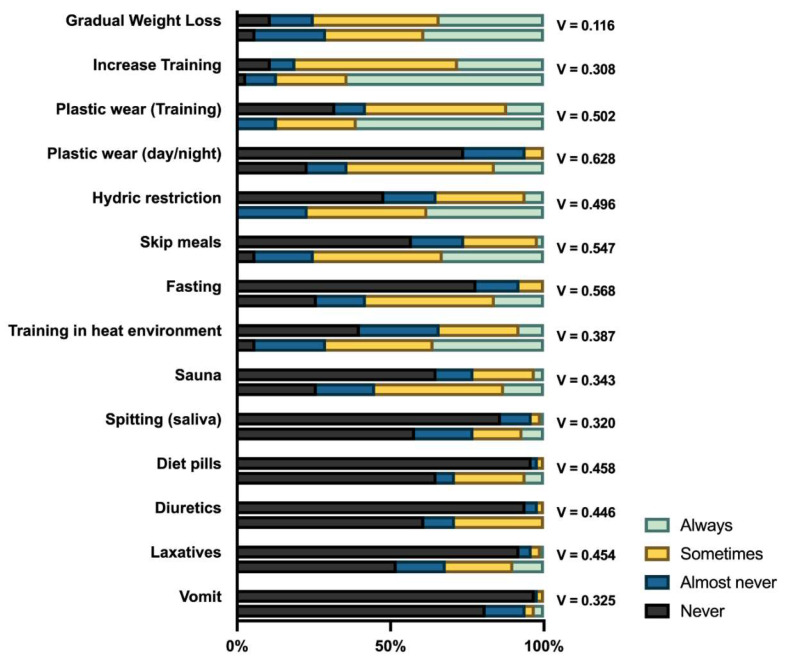
Frequency of use of the different weight-cutting strategies in Cluster 1 (upper bars) and Cluster 2 (lower bars), for each weight-cutting method. Association strength is reported as Cramer’s V.

**Figure 3 ijerph-17-08727-f003:**
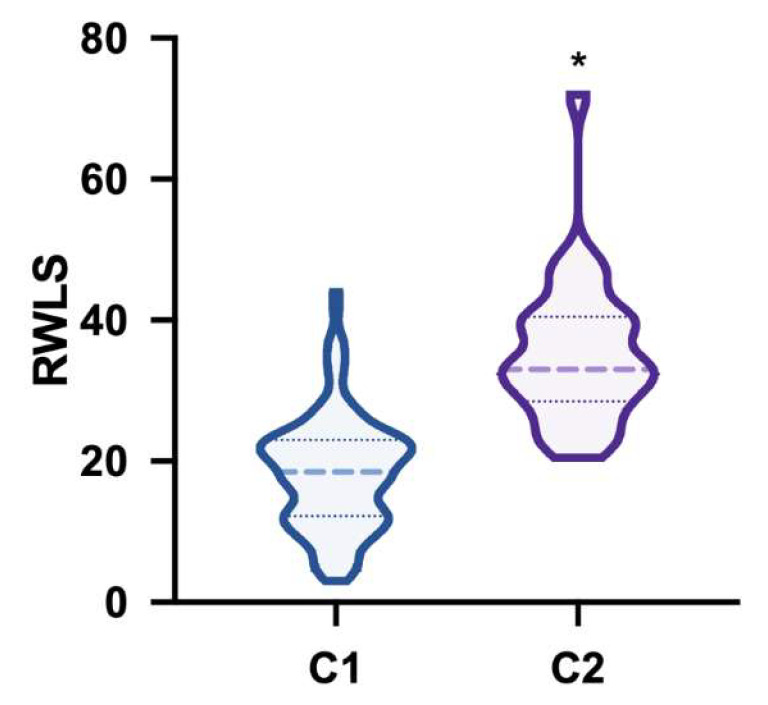
Rapid Weight Loss Score (RWLS) in the two clusters (C1 and C2). The thick line represents the median, while the thin ones represent first and third quartiles; Asterisk indicate statistically significant differences.

**Table 1 ijerph-17-08727-t001:** Participants characteristics. Data are reported as means ± standard deviations. For asymmetric values, median (min–max) have been used. Where applicable, absolute frequencies (percentage) are reported.

	Males (*n* = 144)	Females (*n* = 20)	Total (*n* = 164)
Height (cm)	175.9 ± 7.1	164.2 ± 7.4	174.5 ± 8.1
Weight (kg)	70.7 ± 11.8	56.1 ± 6.1	68.9 ± 12.2
Age (years old)	23.4 ± 6.4	19.2 ± 5.5	22.9 ± 6.4
Education Level			
*Primary School*	24 (16.7%)	5 (25.0%)	29 (17.7%)
*High School*	96 (66.7%)	9 (45.0%)	105 (64.0%)
*Bachelor Degree or >*	24 (16.7%)	6 (30.0%)	30 (18.3%)
Age Started Boxing	16.4 ± 4.9	14.4 ± 4.4	16.2 ± 4.9
Age Started Competing	18.4 ± 4.9	16.8 ± 4.4	18.2 ± 4.9
Training Sessions (*n*/week)	5.9 ± 2.5	6.2 ± 2.3	6.0 ± 2.5
Training volume (h/week)	11.2 ± 7.3	10.5 ± 3.2	11.2 ± 6.9
Boxing Category			
*Amateur (Olympic-* *S* *tyle* *B* *oxer)*	126 (87.5%)	19 (95.0%)	145 (88.4%)
*Professional*	18 (12.5%)	1 (5.0%)	19 (11.6%)
Boxer Level			
*Highest* *C* *alibre*	22 (15.3%)	7 (35.0%)	29 (17.7%)
*Moderate* *C* *alibre*	27 (18.8%)	9 (45.0%)	36 (22.0%)
*Lesser* *C* *alibre*	95 (66.0%)	4 (20.0%)	99 (60.4%)

**Table 2 ijerph-17-08727-t002:** Subjects responses to weight loss specific questions in the two clusters. Data are reported as mean (min–max).

Weight-Cutting History	C1 (*n* = 133)	C2 (*n* = 31)
Most weight lost for a competition (kg)	4.5 (0–17)	6.4 (2–20) *
Number of times of weight cutting in the previous season (times)	1.6 (0–11)	2.5 (0–9) *
Weight usually lost for a competition (kg)	1.5 (0–8)	2.6 (0–10) *
Number of days over which weight is usually lost (days)	12.1 (0–40)	17.2 (2–60)
Age at which began to cut weight for competitions (years)	18.5 (11–34)	17.8 (11–28)
Weight typically regained in the week after the competition (kg)	2.0 (0–10)	4.2 (2–15) *

* *p* < 0.05 (Mann–Whitney with false discovery rate correction).

**Table 3 ijerph-17-08727-t003:** Socio-demographic differences between the two clusters. Data are reported as *n* (% within-cluster).

	Cluster 1 (*n* = 133)	Cluster 2 (*n* = 31)	*p*(χ^2^); Cramer’s V
Age			
*<18 years old*	37 (27.8%)	11 (35.5%)	*p* = 0.398; V = 0.066
*>18 years old*	96 (72.2%)	20 (64.5%)	
Gender			
*Males*	120 (90.2%)	24 (77.4%) *	*p* = 0.05; V = 0.153
*Females*	13 (9.8%)	7 (22.6%) *	
Education Level			
*Primary School*	23 (17.3%)	6 (19.4%)	*p* = 0.923; V = 0.31
*High School*	85 (63.9%)	20 (64.5%)	
*Bachelor Degree or >*	25 (18.8%)	5 (16.1%)	
Category			
*Amateur (Olympic-style)*	122 (91.7%)	23 (74.2%) *	*p* = 0.006; V = 0.215
*Professional*	11 (8.3%)	8 (25.8%) *	
Boxer Level			
*Highest* *C* *alibre*	19 (14.3%)	10 (32.3%) *	*p* = 0.048; V = 0.192
*Moderate* *C* *alibre*	29 (21.8%)	7 (22.6%)	
*Lesser* *C* *alibre*	85 (63.9%)	14 (45.2%)	
People Who May Have Influence:			
Physician/Nutritionist			*p* = 0.007; V = 0.247
*No/Small Influence*	69 (51.9%)	8 (25.8%) *	
*Moderate Influence*	12 (9.0%)	8 (25.8%) *	
*Big Influence*	52 (39.1%)	15 (48.4%)	
Coach			*p* = 0.609; V = 0.078
*No/Small Influence*	33 (24.8%)	8 (25.8%)	
*Moderate Influence*	17 (12.8%)	2 (6.5%)	
*Big Influence*	83 (62.4%)	21 (67.7%)	
Physical Trainer			*p* = 0.171; V = 0.147
*No/Small Influence*	60 (45.1%)	14 (45.2%)	
*Moderate Influence*	13 (9.8%)	0 (0.0%)	
*Big Influence*	60 (45.1%)	17 (54.8%)	
Family			*p* = 0.845; V = 0.045
*No/Small Influence*	86 (64.7%)	19 (61.3%)	
*Moderate Influence*	9 (6.8%)	3 (9.7%)	
*Big Influence*	38 (28.6%)	9 (29.0%)	
Other Boxers			*p* = 0.689; V = 0.067
*No/Small Influence*	89 (66.9%)	21 (67.7%)	
*Moderate Influence*	15 (11.3%)	2 (6.5%)	
*Big Influence*	29 (21.8%)	8 (25.8%)	

* stars indicate rows whose column percentages are significantly different from each other.

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
