# Peer review of "Factors Influencing Weight Loss Practices in Italian Boxers: A Cluster Analysis"

_ijerph, 2020, doi:10.3390/ijerph17238727_

Round 1

Reviewer 1 Report

The manuscript addresses current and relevant issues associated to weight loss strategies used by boxers. There is a high component of originality in the study. The theoretical basis is well founded and justifies the study. The objectives are clearly defined. The methods used are sufficiently described, appropriate and consistent with the study proposal. The results are presented in an attractive manner, and the discussion of the findings show good prospects to offer important contributions to the area of knowledge, which enhances the quality of the manuscript. The bibliographic references are current and pertinent to the theme of study. However, I believe that it is necessary to indicate clearly in the manuscript: (a) the potential limitations and strengths of the study; and (b) the novelty / contribution of the findings of the present study to the improvement of the practical action of coaches, physical trainer, physicians, and nutritionists.

Author Response

Dear Reviewer,

Thank you very much for providing us with comments and suggestions which have helped to improve our manuscript and allowed our results to be presented more clearly. Please find below our detailed responses and attached the revised manuscript.

Point 1: The manuscript addresses current and relevant issues associated to weight loss strategies used by boxers. There is a high component of originality in the study. The theoretical basis is well founded and justifies the study. The objectives are clearly defined. The methods used are sufficiently described, appropriate and consistent with the study proposal. The results are presented in an attractive manner, and the discussion of the findings show good prospects to offer important contributions to the area of knowledge, which enhances the quality of the manuscript. The bibliographic references are current and pertinent to the theme of study. However, I believe that it is necessary to indicate clearly in the manuscript: (a) the potential limitations and strengths of the study; and (b) the novelty / contribution of the findings of the present study to the improvement of the practical action of coaches, physical trainer, physicians, and nutritionists.

Answer: We thank the Reviewer for his effort in reviewing our manuscript, helping us to improve its quality. A section reporting the potential limitations of the study, and its contribution for the improvements of practical actions, has been added in the text (at the end of the Discussion section), as suggested.

Reviewer 2 Report

Authors present the article title: "Factors influencing weight loss practices in Italian boxers: A cluster analysis".It is an important issue of studying due to an uncontrolled weight loss can give as result dangerous health situations. However, there are several aspects that must be taking in account.

Authors

Introduction

Authors must include a description of the different adverse consequences on health.

Methods and results

Although, authors explain that the selection of Cluster was based on statistical results, it is not clear what conditions define each one due to it was not described on the results section. 

Observing the figure 1, it seems to be more recommended a study divide by three Clusters. Perhaps, it would make that grouping of participants included in each cluster will be more homogenous. So, the authors must be done different studies changing the number of clusters selected to analyze which of them explain, not only stadisticantly, the possible situations.

The big part of the differences found in the table 3 are due to the higer number of participants in the Cluster 1.

On the other hand, there are a higher number of amateur compared with professional, and also, very low number of women if is compared with men. Therefore, the main cluster analysis shoud be on the amateur group, and in a subsequently analysis comaparing with professional o women.

The "n" or paticipants showed in the table 2 is different than the table 3.

The discussion should be justified in a more scientific way.

Reviewer 3 Report

Abstract - provided a clear summary of the study

Introduction - well-described the background to the study; the research gap has been clearly articulated

Methods

Section 2.1 - As the number of respondents is average, do provide sample size calculation to show that the study has sufficient power. 

Line 82-83 - I am concerned with the inclusion of <18yr olds in the analysis. Any evidence age could have influenced the weight loss practice?

Line 92 - ..in others studies [16] - please provide few more references

Line 95 - Full list of the collected socio-demographic variables is 95 presented in Table 1. - can be omitted

Results

Provide the response rate to the survey invitation.

The data analysis and presentation are otherwise, well-done.

Discussion

Add a brief paragraph on study limitations.

Author Response

Dear Reviewer,

Thank you very much for providing us with comments and suggestions which have helped to improve our manuscript and allowed our results to be presented more clearly. Please find below our detailed responses and attached the revised  manuscript.

Point 1:

Abstract - provided a clear summary of the study

Introduction - well-described the background to the study; the research gap has been clearly articulated

Answer: Thanks for these comments.

Point 2:

Methods: Section 2.1 - As the number of respondents is average, do provide sample size calculation to show that the study has sufficient power. 

Answer: To calculate the sample size, even if the study is observational, it would be necessary to define a priori a minimum relevant difference between the expected values of a variable relative to two (or more) samples, this defined as a 'primary outcome' of the study. However, since this was an exploratory study and we were not expecting the presence of homogeneous subgroups (i.e. clusters) in our sample, we cannot hypothesise a relevant difference to be used for the sample size calculation. As the number of groups and the relative size derived from the cluster analysis, it was not possible to calculate a priori the sample size. However, in Tab 3 are reported the effect size (as Cramer’s V) of each comparison between clusters: there are no medium-high effects that are not significant, so it is possible to say that the study is not undersized.

Point 3: Line 82-83 - I am concerned with the inclusion of <18yr olds in the analysis. Any evidence age could have influenced the weight loss practice?

Answer: As reported in the Discussion section “It should be noted the absence of significant differences in the presence of under-18 athletes in the two clusters. This means that a good proportion of the youth athletes (about 22%) pertaining to Cluster 2 apply the same dangerous weight-loss practices as adults”. Given this result, we exclude that age influenced weight loss practices, at least in our sample. However, the literature is completely lacking in this regard, and our study highlight the need of further studies on this population.

Point 4: Line 92 - ..in others studies [16] - please provide few more references

Answer: A few more references [3][17] have been added as requested.

Point 5: Line 95 - Full list of the collected socio-demographic variables is 95 presented in Table 1. - can be omitted

Answer: As suggested, the sentence “Full list of the collected socio-demographic variables is presented in Table 1.” has been removed.

Point 6:

Results: Provide the response rate to the survey invitation.

Answer: We cannot answer with precision to this point, as the Italian Boxing Federation, through the regional managers and coaches, helped us to spread the survey among the clubs. As some weight-cutting procedures may be non-legal, or socially accepted, athletes may be reluctant to respond to a direct interview; so, we used a snowball sampling, a technique often used in population which are difficult for researchers to access. Thanks to this sampling methodology, It is possible for the researchers to include people in the survey that they would not have known, through the use of social network. So we did not know exactly to how many people saw the survey but decided not to take part. This point has been clarified in the text in the relevant section 2.2.

Point 7: The data analysis and presentation are otherwise, well-done.

Answer: Thanks for this comment.

Point 8:

Discussion: Add a brief paragraph on study limitations.

Answer: A brief paragraph with the study limitations has been added at the end of the Discussion section.

Round 2

Reviewer 2 Report

Authors respond to the questions suggested.